# Comparing the efficiency of paper-based and electronic data capture during face-to-face interviews

**Alissa Tate** *, **Claire Smallwood**

Department of Primary Industries and Regional Development, Western Australian Fisheries and Marine Research Laboratories, North Beach, Western Australia, Australia

* alissa.tate@dpird.wa.gov.au

## Abstract

On-site surveys involving face-to-face interviews are implemented globally across many scientific disciplines. Incorporating new technologies into such surveys by using electronic devices is becoming more common and is widely viewed to be more cost-effective and accurate. However, Electronic Data Capture methods (EDC) when compared to traditional Paper-based Data Capture (PDC) are often implemented without proper evaluation of any changes in efficiency, especially from surveys in coastal and marine environments. A roving creel survey of recreational shore-based fishers in Western Australia in 2019 enabled a direct comparison between the two methods. Randomisation strategies were employed to ensure biases in using each technique were minimised. A total of 1,068 interviews with recreational fishers were undertaken with a total error rate of 5.1% (CI95%: 4.8–5.3%) for PDC and 3.1% (CI95%: 2.9–3.3%) for EDC. These results confirmed that EDC can reduce errors whilst increasing efficiency and decreasing cost, although some aspects of this platform could be improved with some streamlining. This study demonstrates how EDC can be successfully implemented in coastal and marine environments without compromising the randomised, stratified nature of a survey and highlights the cost-effectiveness of this method. Such findings can be widely applied to any discipline which uses face-to-face interviews for data collection.

## 1 Introduction

Efficient and accurate data recording is essential in all scientific research and monitoring programs to ensure quality and cost-effectiveness while maintaining confidence in research outputs. Paper-based Data Capture (PDC) has historically been the main method used to record information. However, in the past decade, hardware (*i.e.*, personal digital assistants, tablets, remotely operated cameras) for implementing Electronic Data Capture (EDC) has become substantially cheaper and more accessible [1]. The number of software programs (*e.g.*, Cybertracker, Survey 123) available for researchers to customise data collection to support specific projects on this hardware has also increased rapidly [2,3]. The perceived benefits of EDC over

made available upon approval to researchers who meet the criteria for access to confidential data (contact via DataControl-SADA@fish.wa.gov.au).

**Funding:** Funding was provided by the Department of Primary Industries and Regional Development.

**Competing interests:** The authors have declared that no competing interests exist.

**Abbreviations:** CAPI, computer assisted personal interviews; EDC, electronic data capture; PAPI, paper and pencil interviews; PDC, paper-based data capture.

PDC include increased efficiency and the provision of more timely data, cost-effectiveness, accuracy, in terms of a reduction in data transcription errors, and facilitation of real-time validation and reporting [4]. The high uptake of EDC by researchers across a wide variety of disciplines in recent years highlights its versatility and ability to provide data at spatial and temporal scales that are not available using human resources due to cost or practical reasons, such as 24-hr day sampling of estuarine shore-based recreational fishing [5] or underwater footage of trawl nets for by-catch mitigation [6].

Remote data collection by electronic mechanisms is now widely used across scientific disciplines. For example, video or still cameras and other sensor systems can be positioned at fixed terrestrial locations or access points such as boat ramps or hiking trails to monitor recreational activity levels across a 24-hr day [5,7–11] while waterproof systems can be used to monitor a suite of species or habitats [4,12] as well as vessel activity [13]. Unmanned aerial vehicles have applications for many projects including monitoring of specific activities, species, and habitats [14–16].

The accessibility and functionality of EDC for on-site surveys have also improved, in the past decade and are increasingly implemented into on-site surveys. Trained field officers can use tablets or PDAs to record data from face-to-face interviews or observational data collected during interviews using randomised or probability-based survey designs [4]. Citizen science projects are also able to take advantage of EDC by developing smartphone apps for volunteers who 'opt-in' to report information relevant to their research, such as catch data from recreational fishers, health monitoring or animal tracking [3,17,18].

A strategic assessment of changes to existing data collection methods is important to understand the benefits, limitations, and potential biases that may occur and the implications this may have for reporting on the findings of a research project. This includes assessing a transition from PDC to EDC in the context of their application in face-to-face interviews. Previous studies have compared EDC with PDC although the focus has been on the efficiency of devices rather than the accuracy of the data collected [19]. [20] explored the application of EDC in health surveys comparing efficiencies of PDC compared to EDC when used by respondents completing a questionnaire. [21,22] also demonstrated how EDC can be used to complete face-to-face interviews comparing the use of tablets and notebooks to PDC, ultimately concluding that EDC was preferred by survey staff, saved time, and reduced survey costs. However, no study has quantified the error rates and undertaken a direct comparison of the accuracy, timeliness, and cost-effectiveness of PDC and EDC for face-to-face interviews.

This study aims to test if EDC is a better option for data collection than PDC during face-to-face interviews. By implementing concurrent PDC and EDC methods during a survey of shore-based recreational fishers it was possible to quantify the differences in error rates across a number of metrics including (1) data accuracy and timeliness, (2) practicality and ease-of-use and (3) cost-effectiveness and reliability. Direct comparison of EDC and PDC has rarely been undertaken and enables careful consideration of the implications of each platform and ensures that data quality and survey design are not being compromised. This comparison is especially useful for understanding the effectiveness of EDC in outdoor, coastal and marine environments, which have additional challenges (*i.e.*, remote from charging stations, exposure to water).

## 2 Methods

### 2.1 Ethics statement

The pilot study reported here was integrated into an existing roving survey of shore-based recreational fishing, and approved by the Department of Primary Industry and Regional

Development, Western Australia (DPIRD), under Commissioner's instruction No.7 –Code of Ethics. The collection of recreational fishing data and comparison of field officer error was guided by principles of informed consent, voluntary participation, confidentiality, and collection of only relevant information. Recreational fishers and field officers provided verbal consent to participate and were aware of the intent to publish from the outset of the survey. Data obtained from these surveys are the property of DPIRD and are not publically available.

## 2.2 Study area

Western Australia has a population of 2.595 million people (ABS, 2018), of which 25% participate in recreational fishing annually [23]. The majority of the population (and fishing activity) occurs in the West Coast bioregion [24], which includes the Perth Metropolitan area which extends along the coast for ~100km (Fig 1). There are numerous platforms for shore-based recreational fishing within this area, including groynes, natural rocky outcrops, intertidal reef platforms, jetties, and sandy beaches. A suite of nearshore finfish species is targeted by shore-based recreational fishers in this area [25,26].

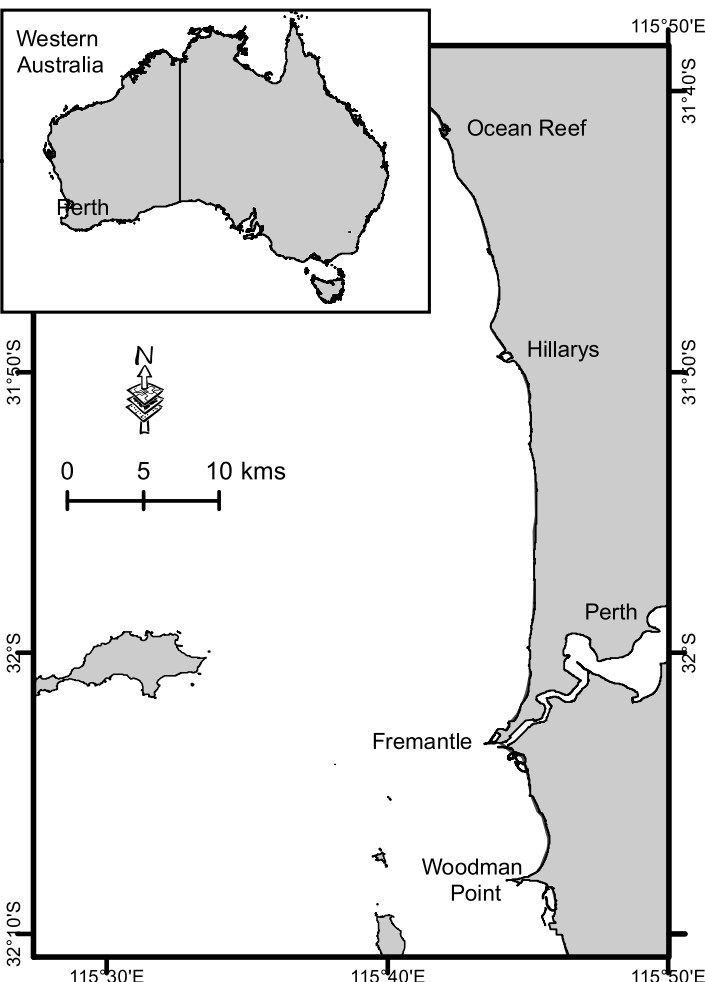

**Fig 1. Perth metropolitan area in western Australia and survey extent (from Ocean Reef to Woodman Point).**

## 2.3 Survey design

A 5-month roving creel survey of recreational shore-based fishers was undertaken in the Perth Metropolitan area between Ocean Reef and Woodman Point in 2010, and then annually from 2014–2019 (Fig 1). The randomised, stratified survey design follows well-documented protocols [27] and has been implemented consistently for all surveys [25,26]. Standardised questionnaires for each survey interview captures 27 fields related to the survey day (i.e. location and environmental conditions), fishing trip (i.e. gear and targeting), catch (i.e. species caught), or measurement (i.e. fish length) (Table 1). Survey days and locations were determined *a priori* from the survey design, while the information obtained for trip and catch were random variables as presented on any given survey day, and if any catch was available, a random sample of fish was measured.

PDC has been used for all surveys from 2010 to 2018. However, in 2019, EDC was introduced alongside the existing PDC method. The hardware used for EDC was an Apple iPad Pro 10.5' (iOS) with FileMaker Pro (FileMaker Pro Advanced Version 18.03.3.317) selected to develop a relational database with forms for data entry. The FileMaker database was designed as an exact replicate of the existing Microsoft Access database which the data from PDC were manually entered into, thereby enabling easy comparison between PDC and EDC.

Every survey was undertaken by two field officers who both recorded all information from the interview using either PDC and EDC. Each field officer was therefore assigned to be one of two roles, namely;

- Interviewer–responsible for conducting the verbal interview with fishers, measuring the catch and recording all information from the interview or,

- Scribe–responsible for recording all information from the interview.

To minimise bias, a roster was generated to ensure that there was a random allocation of roles (Interviewer or Scribe) and data collection platform (PDC or EDC) for each staff member at each location within a survey. In total, the survey team consisted of five survey staff, four of which had no previous experience with this survey or prior training for either technique. All staff were instructed not to corroborate or cross-check between PDC and EDC methods to maintain independence, similar to strategies implemented during other paired-observer surveys [28,29].

**Table 1. Description of four interview sections and the number of possible errors.**

| Section number | Section name | Variables collected | Number of fields | Number of records | Number of possible errors |
|---|---|---|---|---|---|
| 1 | Survey | Date, location, interviewer, field officer role, weather conditions (wind direction, wind speed, rainfall, cloud cover), beach type arrival time, departure time, count of people fishing on arrival, count of people fishing on departure | 14 | 988 | 13,832 |
| 2 | Trip | Fishing platform, time, postcode of residence, group size, number actively fishing, gear type, number of gear used, time spent fishing, avidity, target species | 9 | 1,068 | 9,612 |
| 3 | Catch | Name of species caught, number of each species kept and released | 3 | 1,068 | 3,204 |
| 4 | Length | Species name and total length | 1 * | 1,468 | 1,468 |
| | | | Total | 4,592 | 28,116 |

* if catch was available, a random sample of fish was measured, therefore 1 record exists for each fish measured.

## 2.4 Analysis

**2.4.1 Accuracy.** The accuracy of the data collected for the two platforms was assessed for all 27 variables collected on the interview form and was broadly defined as two types of inaccuracy; (1) missing and (2) error. Missing referred to when data was not entered (*i.e.*, the variable was left blank), while error referred to when incorrect data was entered during the interview, which was identified either during the QA/QC process or a data mismatch between the two survey modes. These measurements were made across all four sections of the interview and were defined as survey error, trip error, catch error, or length error (Table 1). Each of these combinations of inaccuracy type (n = 2) and interview sections (n = 4) were considered for each role of the field officer as interviewer or scribe (n = 2).

The approach taken to identify and quantify the accuracy and errors in interviews was consistent throughout the survey and required several steps to enable a comparison between the two survey modes (Table 2). After each survey was completed the paper sheets were returned to the office for data entry while the electronic data were synced with the database. The second step in the process only concerned PDC as the datasheets needed to be first validated manually for errors by data entry staff before entry into the database. This is a standard procedure for PDC in all years that this study has been completed. As described above, each measure of inaccuracy type (n = 2) was quantified by the different sections of the interview (n = 4) and field staff roles (n = 2).

The next step was to merge the PDC and EDC datasets to allow a comparison of each field for inconsistencies. Variables that did not match were marked and investigated to identify the inaccuracy type (missing, error) and assign it to the field staff role (interviewer or scribe). In the case of PDC, the inaccuracy type could also be attributed to the data entry officer. If there was no evidence as to where the data mismatch originated, then the interviewer was considered the most accurate record as they were the closest party and most present in the interview.

At the end of this process, it was possible to provide an overall error rate for each survey mode as a proportion of the total number of possible errors (Table 1). Within this overall error rate, it was also possible to investigate where the highest proportions of errors occurred (*i.e.*, in which section of the interview and for which field officer role). Where possible, 95% Confidence Intervals (95%CI) were calculated using a binomial test for proportions to enable statistical comparison of data.

**2.4.2 Practicality.** To assess EDC with regards to practicality (*i.e.*, ease of use and speed) when compared to PDC a survey of field staff was conducted one month after the field survey commenced. The survey asked seven questions each with multiple choice answer (Always, Usually, Sometimes, Rarely, Never) relating to (1) reliability of the device (*i.e.*, log-in, battery life, connection), (2) durability (*i.e.*, able to withstand wet hands, sand, weather conditions),

**Table 2. Outline of the approach implemented to count and compare errors between Paper-based data capture and electronic data capture survey modes.**

| Step | Paper-Based Data Capture (PDC) | Electronic Data Capture (EDC) |
|---|---|---|
| 1 | Datasheets returned post-survey | Data synced to FileMaker database |
| 2 | Manual validation of datasheets by two people and count of inaccuracy type (missing, error), interview section (survey, trip, catch and length) and field officer role (interviewer, scribe) | NA |
| 3 | Data sets merged and data mismatches identified between survey modes for every variable collected by inaccuracy type (missing, error), interview section (survey, trip, catch, and length), and field officer role (interviewer, scribe). Note: for PDC only non-field officer errors were classified as data entry errors | |

(3) reliability of the application (*i.e.*, log-in, connection issues, syncing), (4) durability of the application (*i.e.*, intuitiveness, ability to make corrections), (5) survey mode preference, (6) difficulty of EDC use, and (7) general comments. The same survey was completed by staff at the end of the survey to determine whether opinions had changed over time after staff had been working regularly with EDC throughout the 6-month survey.

**2.4.3 Cost-effectiveness.** The cost-effectiveness of each survey mode was considered broadly in terms of the equipment and labour (in terms of time and cost).

## 3 Results

### 3.1 Accuracy

Sampling occurred on 60 days between February and June 2019 at a total of 988 survey locations. The 'survey' section of the interview was completed for each of location visited and documented date, time, location, and weather information, as well as counts of fishers. There were 1,068 face-to-face interviews with recreational shore-based fishers during the entire survey period to document the fishing trip and catch for each interview. A random sample of 1,468 length measurements of the retained catch was also completed (Table 1).

Data entry errors occurred on 208 occasions (9.0% of recorded errors) during PDC for the survey period, with most occurring whilst entering fishing trip data (section 2). These errors in data entry were consistent between February and May and decreased in June (Fig 2). As these errors are specific to PDC they were not considered in the comparison of accuracy and error rates between the two survey modes.

The total number of inaccuracies identified across all interviews and variables was 1,433 (5.1%, CI95%: 4.8–5.3%) of all possible errors for PDC and 897 (3.1%, CI95%: 2.9–3.3%) for EDC. These error rates differed across the survey period with an average of 7.8 errors per day (CI95%: 7.1–8.4) in February (the first survey month), peaking in May (9.6 errors per day,

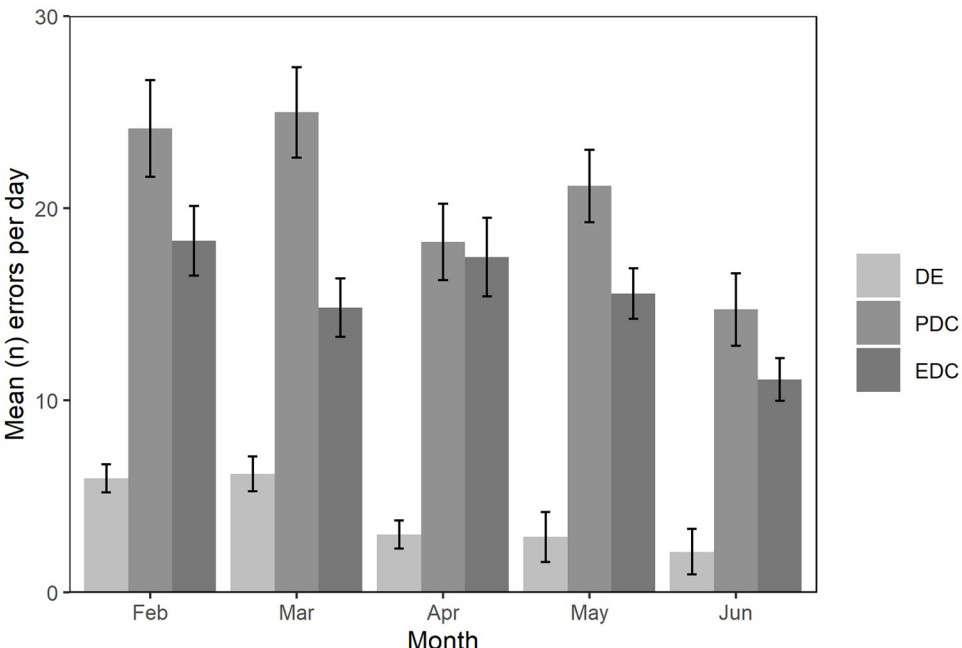

**Fig 2. Mean error rate per month (95% CI) occurring for each survey mode (EDC, PDC) and for data entry errors (DE, which relate to PDC only).**

**Table 3. Summary of inaccuracy type (missing, error) occurring in each survey mode (PDC, EDC) by each section of the face-to-face interview (survey, trip, catch and length) and field officer role (interviewer, scribe).**

| Paper-based Data Capture | | | | |
|---|---|---|---|---|
| **Field officer role** | **Section** | **Error** | **Missing** | **Total** |
| Interview | Survey | 89 | 66 | 155 |
| | Trip | 138 | 43 | 181 |
| | Catch | 57 | 12 | 69 |
| | Length | 160 | 35 | 195 |
| | Total | **444 (74%)** | **156 (26%)** | **600 (49%)** |
| Scribe | Survey | 58 | 50 | 108 |
| | Trip | 113 | 28 | 141 |
| | Catch | 22 | 3 | 25 |
| | Length | 22 | 329 | 351 |
| | Total | **215 (34%)** | **410 (65%)** | **625 (51%)** |
| **Total** | | | | **1225 (53%)** |
| **Data Entry** | | | | **208 (9%)** |
| Electronic Data Capture | | | | |
| **Field officer role** | **Section** | **Error** | **Missing** | **Total** |
| Interview | Survey | 37 | 18 | 55 |
| | Trip | 45 | 12 | 57 |
| | Catch | 10 | 33 | 43 |
| | Length | 29 | 290 | 319 |
| | Total | **121 (26%)** | **353 (74%)** | **474 (53%)** |
| Scribe | Survey | 87 | 13 | 100 |
| | Trip | 137 | 12 | 149 |
| | Catch | 53 | 35 | 88 |
| | Length | 38 | 48 | 86 |
| | Total | **315 (74%)** | **108 (26%)** | **423 (47%)** |
| **Total** | | | | **897 (38%)** |

Note: Bold numerals indicates a column sub-total or total.

CI95%: 8.7–10.5) and lowest in June (6.4 errors per day, CI95%: 5.5–6.5) than June (the last month of the survey) (Fig 2). The patterns across months was consistent for PDC and EDC.

A more detailed breakdown of the type of inaccuracy (error, missing), and the interview section in which it occurred, as well as the effect of the field officer role on the error rate revealed some patterns between PDC and EDC (Table 3). The role of the field officer made little difference to the error rate when using the PDC method (interviewer had 49%, CI95%: 46–52% of errors compared to the scribe 51%, CI95%: 48–54%) but, had more of an influence on the EDC, with the interviewer having slightly more inaccuracies than the scribe (53%, CI95%: 49–56% vs 47%, CI95%: 44–50%). However, when considering the field officer role within each survey mode there was no consistent pattern in error types.

Error rates were more frequent in the length section of the interview for both the interviewer and scribe in PDC as well as for the interviewer in EDC. However, for scribes in the EDC, the majority of errors were in the trip section (Table 3). The majority of inaccuracies when using PDC were errors *i.e.* data entered incorrectly onto the datasheet, and it was missing fields that were the highest recorded inaccuracy for EDC most of which came from the catch and length sections where the field officer role was interviewer (Table 3, Fig 3).

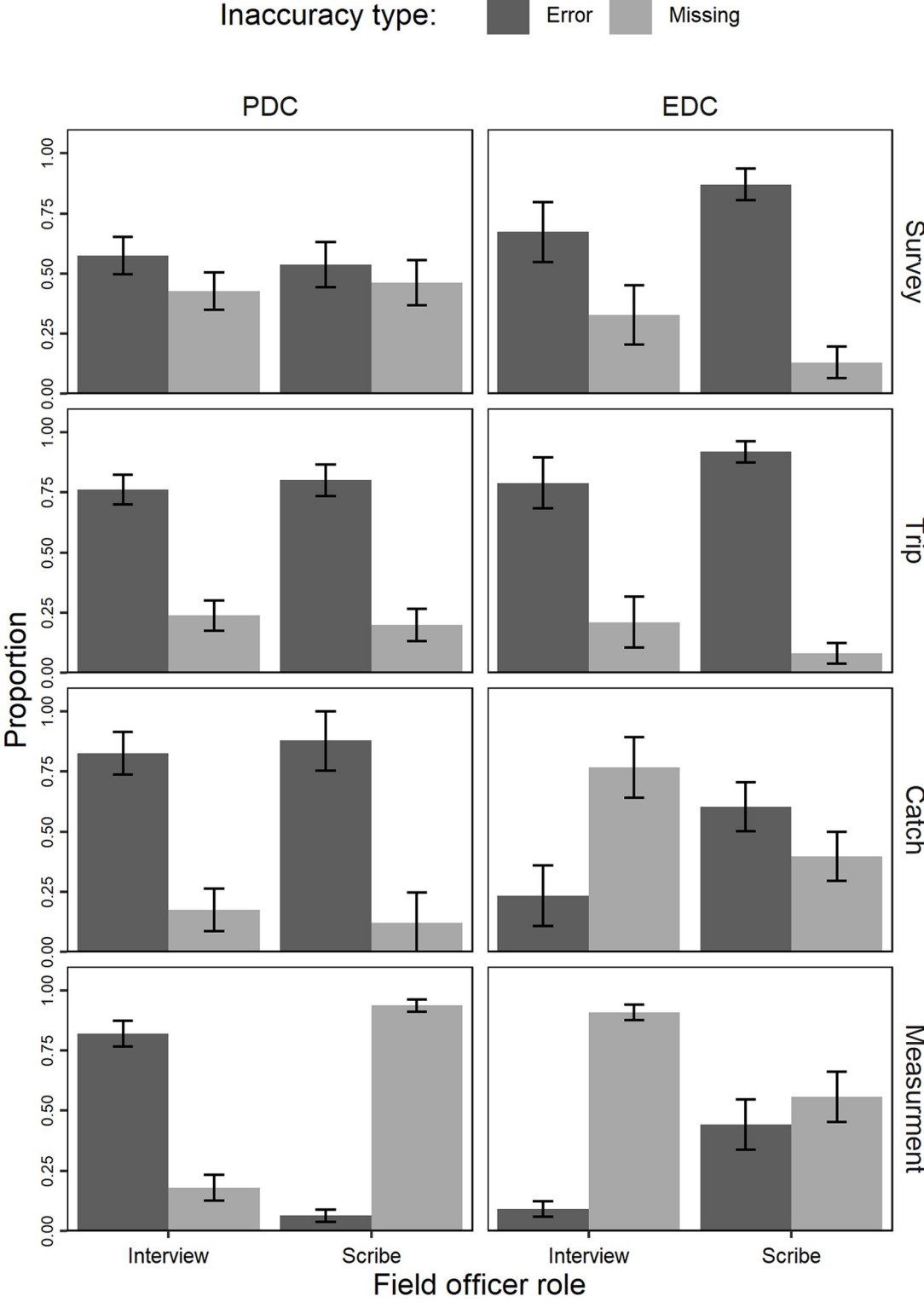

**Fig 3. Proportion of errors occurring within each section of the face-to-face interview by inaccuracy type (error, missing) and field officer role (interviewer, scribe).** Each plot showing the proportion of each inaccuracy type (error, missing) made by the Interviewer and Scribe when using PDC and EDC.

The timeliness of the data being available for analysis was also compared between PDC and EDC. Data recorded using EDC was available as soon as the tablet was synced to the FileMaker database. Although this was often done instantaneously, occasional difficulties with internet connections meant this syncing process was delayed until the end of the survey day when field officers returned to the office. Data recorded using PDC was returned to the office on the day of the survey and was entered into the database within 5 working days.

### 3.2 Practicality

Although the sample size (n = 5) was small, the results from the staff survey completed after the first month and again at the completion of the survey provided some insights into the practicality of PDC when compared to EDC in several areas. Overall, the opinions of the staff improved throughout the survey, changing from 'sometimes' to 'usually' for most questions, which reflected the consensus that the device got easier to use with time and experience. Initially, the paper surveys were perceived to be easier to manipulate, to add or change data. However, after the first staff survey, measures were taken to improve the functionality both the application and the device. In the general comments of the survey, staff indicated that these improvements assisted them greatly and improved the overall functionality of the device however, further change to the layout would improve the device application.

### 3.3 Cost-effectiveness

There was no observed difference in the time required to collect data using PDC and EDC and, except for data entry, costs were similar. No data was lost from either method throughout the study. The return of data for EDC was instantaneous and allowed for immediate review, identification, and correction of mistakes, where PDC took 1–2 days (depending on field staff proximity to the office) to complete the same process. Data entry, which was only required for PDC, took a total of 180 hours (~3 hrs per survey) to enter, including validation time.

## 4 Discussion

A survey of recreational shore-based fishers in Western Australia provided a rare, direct comparison of PDC and EDC incorporated into face-to-face interviews as part of randomised, stratified survey design. The findings revealed that although some aspects of EDC could be improved to streamline data collection, it does provide several benefits when compared to PDC including improvements in accuracy, timeliness of data availability, practicality, and cost-effectiveness. Face-to-face interviews are an important data collection technique used across many scientific disciplines. However, the rapid development and accessibility of emerging technologies often result in their incorporation into surveys without appropriate testing or comparison with existing methods which may introduce biases into the data. This study is, therefore, an important contribution to the growing literature on this topic.

The error rate for EDC (3.1%, CI95%: 2.9–3.3%) was lower than PDC (5.1%, CI95%: 4.8–5.3%). These errors fluctuated across the survey period for both survey modes, although this was more evident for PDC. This trend indicates that staff errors reduced over time as they became more proficient in data collection, but also highlights the need for ongoing training to identify issues early and provide additional reinforcement of data collection practices. In addition, as the study period for this trial was relatively short there is no reference to gauge the effect staff complacency to EDC overtime. If EDC is implemented in future surveys periodic corroborative studies would assist in ensuring the data quality remains unchanged.

The majority of errors for both PDC and EDC occurred during the length section of the interview, which involved measuring a random sample of the retained catch. This is likely due

to measuring catch being one of the most involved activities during the interview (*i.e.*, handling wet and sometimes active fish) which is challenging for both survey modes if an operators' hands are wet and soiled when entering data. This is an issue not often faced by other studies trialling EDC, but one that is significant to fisheries sampling. In other disciplines, where data is collected by individuals going door-to-door and seldom collecting biological data, EDC improved both the timeliness and efficiency of data collection [20,21]. The real test for EDC in on-site fisheries sampling is its adaptability to difficult environmental conditions, an issue easily mitigated with additional staff but the potential for increased errors may need to be considered for surveys that are only completed by a single field officer.

Results showed the timeliness of data delivery was improved in EDC, and data collected using PDC had a small 0.7% of additional error as well as a lag of 5 working days which, although still relatively efficient, was still an added cost in terms of labour. The increased time and resources (additional data entry staff) required when using PDC has previously been acknowledged, as has the need for prompt data collection to ensure errors are identified and mitigated in the early stages of a project, and analysis and reporting keeps pace with rapid changing fisheries [30]. The timeliness of EDC allowed for the prompt delivery of data through instantaneous syncing with the database, ultimately improving the accuracy of data collected.

Two staff surveys were completed to help assess EDC throughout the study. Staff initially preferred PDC compared to EDC across most elements of reliability and durability of both the device and application. This was due to PDC allowing for more flexibility in recording data, as staff could partially fill out the form during the interview and then complete immediately after the interaction, similar to found by [21]. This flexibility in data entry was not overly important for this survey which had short fields to complete, however for studies where open-ended, opinion based questions are asked the flexibility of PDC may be more efficient. By the completion of the study, all staff preferred EDC. After a period of adjustment adapting to the new technology, feedback from the initial staff survey was used to modify the device and streamline the application to improve its efficiency and ease-of-use. Modifications included the implementation of additional data control checks in the application (*i.e.*, improving search functionality on drop-down lists and ensuring all fields were required or could be automated where possible, such as the date and time of interview), and a waterproof cover and shoulder strap to protect equipment when conducting face-to-face interviews in hard-to-reach places (*i.e.*, rock platforms).

The device proved durable for the length of the survey (7 hours), with no major disruptions or data collection issues due to battery life. This case study was completed in a densely populated area with high levels of mobile coverage and, as a result, there were few issues experienced with instantaneous synchronisation of the tablet with the database, and no loss of data or battery life occurred. However, many study areas are remote and mobile coverage or charging stations may not be readily available. The device and application can store and backup data until a connection can be obtained, which goes some way to mitigate this issue, although a local backup on a hard drive may be appropriate if unable to connect for longer periods.

EDC offers advantages to collecting scientific data across a range of disciplines and survey methods. This study highlights many benefits, however, several lessons learned during the transition from PDC to EDC have wide application to other research. Firstly, it is important to assign adequate lead-in time for the selection of appropriate software and its development, including time for testing to allow for modifications. Serious consideration also needs to be given to how data collected via EDC aligns with existing databases as well as electronic security measures in place within an organisation. The amalgamation of EDC with existing long-term databases was an ongoing issue in this study and, although addressed in the short-term, will ultimately require a new custom-made database for EDC to be more efficiently incorporated

into future surveys. As a case study with only one survey occurring at a time, it was only necessary to use a single tablet device. Currently, EDC is being incorporated on a larger survey of recreational fishers with multiple tablet devices which is effectively syncing data from multiple locations at the same time. The spatial and temporal scales of application for EDC are therefore also important to consider.

The application of EDC in face-to-face interviews have several benefits, including increased efficiency, the provision of more timely data, increased cost-effectiveness, as well as increased data quality [1,4,21]. However, many EDC techniques such as smartphone applications are based on 'opt-in' samples which, depending on the objective of a survey, present challenges for statistical estimation because those people who self-report may not be representative of the target population [17]. This case study has demonstrated that, even in the context of face-to-face interviews undertaken as part of a randomised, probability-based survey, EDC provides a practical alternative to PDC; improving the accuracy of data and timeliness of data availability while having improved cost-effectiveness. These findings provide a quantification of how EDC compares to PDC across a number of metrics and provides confidence for transitioning between these survey modes in any discipline implementing face-to-face interviews.

## Acknowledgments

The authors would like to thank the Department of Primary Industries and Regional Development (DPIRD) for planning and implementation of this survey as well as the field interviewers and data entry staff. We thank DPIRD staff Rachel Marks and Stuart Blight for their insights and technical assistance, and Karina Ryan and Stephen Taylor for their valuable comments. We are extremely grateful to all the recreational fishers who participated in the surveys and to the reviewers and the editor for their insightful comments.

## Author Contributions

**Conceptualization:** Alissa Tate, Claire Smallwood.

**Data curation:** Alissa Tate.

**Formal analysis:** Alissa Tate, Claire Smallwood.

**Investigation:** Alissa Tate.

**Methodology:** Alissa Tate, Claire Smallwood.

**Project administration:** Alissa Tate.

**Resources:** Alissa Tate.

**Supervision:** Claire Smallwood.

**Validation:** Alissa Tate, Claire Smallwood.

**Visualization:** Alissa Tate.

**Writing – original draft:** Alissa Tate.

**Writing – review & editing:** Alissa Tate, Claire Smallwood.

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
