## [Decision Letter · Decision Letter 0]

30 Nov 2020

PONE-D-20-28545

Comparing the efficiency of paper-based and electronic data capture during face-to-face interviews

PLOS ONE

Dear Dr. Tate,

Thank you for submitting your manuscript to PLOS ONE. After careful consideration, we feel that it has merit but does not fully meet PLOS ONE’s publication criteria as it currently stands. Therefore, we invite you to submit a revised version of the manuscript that addresses the points raised during the review process.

We look forward to receiving your revised manuscript.

Kind regards,

Dimitrios K Moutopoulos, PhD

Academic Editor

PLOS ONE

Additional Editor Comments:

I consider the basic premise, method developed and questions asked by this study as valuable and interesting. This issue is surely of great interest and the lack of information about the on-site surveys of recreational fishery, compromises the capacity to put in place effective policies, both in terms of stock assessment and fisheries management strategies. The strength of the study is the novelty and broad usefulness of this approach. This task is not easy, with very different levels of data availability, especially for the data-poor areas and high heterogeneity in the quality of data between the different fishery compartments, especially for the recreational fishery, which exhibited very different levels of data availability and importance between countries. These data constraints might be usefully to be mentioned in the discussion.

The survey design and data analysis are clear and well explained.

However, there are certain issues that need to be addressed before publication. More specifically, the authors should clarify the terminology of certain aspects of errors (e.g. accuracy, efficiency). For consistency reasons error rate’ may confuse the reader in suggesting it only refers to errors, and not missing data. Title should be also reconsidered. In the results, authors should clarify the sampling strategy.

Also, which are the geographical constraints that should be taken into account to make the case study expand in broader spatial magnitude in order to be a controllable and well-defined example?

A table is missing from the submitted manuscript (in line 222 Table 5 was wrongly mentioned instead of Table 4) and the authors should be consistent with the journal guidelines especially in the cross-reference with the figure (e.g. Fig. 1, Fig 3, Figure 2). In Table 3 it might be useful for the audience to have a clear picture of the results by showing totals for error and missing data (ie., combining both interviewer and scribe).

More comments are raised by the reviewer comments.

Journal Requirements:

"The authors would like to thank the Department of Primary Industries and Regional Development (DPIRD) for funding, planning, and implementation of this survey as well as the field interviewers and data entry staff."

3. We note that Figure 1 in your submission contains map images which may be copyrighted. All PLOS content is published under the Creative Commons Attribution License (CC BY 4.0), which means that the manuscript, images, and Supporting Information files will be freely available online, and any third party is permitted to access, download, copy, distribute, and use these materials in any way, even commercially, with proper attribution. For these reasons, we cannot publish previously copyrighted maps or satellite images created using proprietary data, such as Google software (Google Maps, Street View, and Earth). For more information, see our copyright guidelines: http://journals.plos.org/plosone/s/licenses-and-copyright.

(1) You may seek permission from the original copyright holder of Figure 1 to publish the content specifically under the CC BY 4.0 license. 

4. Please ensure that you refer to Figures 2 and 3 in your text as, if accepted, production will need this reference to link the reader to the figure.

5. Please include a copy of Table 5 which you refer to in your text on page 12.

Reviewers' comments:

Reviewer's Responses to Questions

**Comments to the Author**

1. Is the manuscript technically sound, and do the data support the conclusions?

Reviewer #1: Yes

2. Has the statistical analysis been performed appropriately and rigorously? 

Reviewer #1: Yes

3. Have the authors made all data underlying the findings in their manuscript fully available?

Reviewer #1: No

4. Is the manuscript presented in an intelligible fashion and written in standard English?

Reviewer #1: Yes

5. Review Comments to the Author

Reviewer #1: This paper deals with comparing efficiency of two types of tools (paper based and electronic) to capture data during face-to-face interviews to recreational shore fishers in Western Australia. Authors measure efficiency of the two methods via three main dimensions: Accuracy, Practicality, and Cost-effectiveness. This is a very interesting and relatively novel area of research, particularly for on-site surveys of recreational fishing. As the authors pointed, it is thus an important contribution to the field; it provides relevant information that can be used to support future decisions on the type of tool used, electronic or paper based, to collect data in similar surveys.

Overall, the paper is very well written, and the various sections are balanced. The Introduction explores relatively well the state of the art for this topic and lays the ground on why this type of research is needed. The review of the previous research on the topic is not extensive, and the number of references included is relatively small, but this can be due the limited number of studies on the topic. The survey design and data analysis are clear and well explained. The statistical analysis used to test the accuracy of the two approaches is relatively simple but seems appropriate and sufficient to explore the data and support the findings. The Results and Discussion also describe clearly and explore well the main findings. There are only a few points which the authors should consider to potentially improve the paper.

Title (lines 1 and 2): If the character limit allows, I would suggest the Title lists exactly what the study investigated: accuracy, practicality and cost-effectiveness. ‘Efficiency’ is somewhat of a vague term in my opinion.

Abstract (lines 18, 19): Consider including here exactly what the direct comparison was about: accuracy, practicality, and cost-effectiveness

Abstract (line 21); ‘Error rate’: Here and elsewhere in the paper (e.g., keywords, lines 188, 201, 203, etc), whenever the authors are referring to the total inaccuracies (missing data and errors) should use the term ‘inaccuracy rate’ rather than ‘Error rate’. ‘Error rate’ may confuse the reader in suggesting it only refers to errors, and not missing data.

Introduction (line 68): For consistency with the terminology used in other parts of the paper, authors should use the term ‘practicality’ rather than ‘timeliness’

Methods, Study area (line 93): This seems to refer to figure 1, but there is an error in the cross-reference to this Fig.

Table 1- The authors should consider replacing the term “errors” by “inaccuracies” in both the table caption and header. As noted above, this may confuse the reader as error is one of the two types of inaccuracies analyzed by the authors.

Results (line 184): This seems to refer to figure 2, but there is an error in the cross-reference to this Fig. Note also that the Figure captions seem to be missing from the submitted PDF.

Results (line 189): Could the authors clarify if each survey day had the exact same number of interviews. If so, than the average number of inaccuracies per day (as noted above, ‘inaccuracies’ seems a better term to use here, as this seems to refer to both missing data and error in data entered) seems appropriate.

If not (ie., if each survey day had different number of completed interviews), than the average number of inaccuracies per filled interview within each month would seem to be a more appropriate metric to use.

Table 3 – the authors should consider adding totals for error and missing data (ie., combining both interviewer and scribe) to Table 3. This would help the reader in interpreting this result.

Results (line 222): There is reference to Table 5 in the text, but no reference to Table 4, and both Tables 4 and 5 seem to be missing from the submitted PDF.

Discussion (line 266): This percentage of 0.7% seems to be mentioned for the first time. The authors should consider including it somewhere in the Results, so the reader clearly understands where this is coming from.

Discussion (line 275): based on the information provided in the Methods, all of the questions or fields included in the survey seemed to be multiple option, or for relative few characters (if a free text field). It would be potentially interesting if the authors could add a note or two on the potential practically of EDC vs PDC in the use of open-ended questions with potentially long responses (e.g., opinion on existing regulations).

6. PLOS authors have the option to publish the peer review history of their article (what does this mean?). If published, this will include your full peer review and any attached files.

Reviewer #1: No

---

## [Author Response · Author response to Decision Letter 0]

1 Feb 2021

Response to reviewer document is attached. Specific responses to each comment can be found in this document.

---

## [Editor Report · Decision Letter 1]

10 Feb 2021

Comparing the accuracy, practicality and cost-effectivenes of paper-based and electronic data capture during face-to-face interviews

PONE-D-20-28545R1

Dear Dr. Tate,

We’re pleased to inform you that your manuscript has been judged scientifically suitable for publication and will be formally accepted for publication once it meets all outstanding technical requirements.

Kind regards,

Dimitrios K Moutopoulos, PhD

Academic Editor

PLOS ONE

Additional Editor Comments (optional):

The authors followed all the comments raised by the reviewers and, particularly regarding the terms of ‘error rate’ and the potential confusion around its meaning. The ms is now coherent and robust, and it can be published as it is.
---

## [Editor Report · Acceptance letter]

15 Feb 2021

PONE-D-20-28545R1 

Comparing the efficiency of paper-based and electronic data capture during face-to-face interviews 

Dear Dr. Tate:

I'm pleased to inform you that your manuscript has been deemed suitable for publication in PLOS ONE. Congratulations! Your manuscript is now with our production department. 

Kind regards, 

on behalf of

Dr. Dimitrios K Moutopoulos 

Academic Editor

PLOS ONE